# Systematic analysis of protein turnover in primary cells

Toby Mathieson [1], Holger Franken[1], Jan Kosinski[2], Nils Kurzawa[3], Nico Zinn[1], Gavain Sweetman[1], Daniel Poeckel[1], Vikram S. Ratnu[3], Maike Schramm[3], Isabelle Becher[3], Michael Steidel[1], Kyung-Min Noh[3], Giovanna Bergamini[1], Martin Beck [2], Marcus Bantscheff [1] & Mikhail M. Savitski[3]

A better understanding of proteostasis in health and disease requires robust methods to determine protein half-lives. Here we improve the precision and accuracy of peptide ion intensity-based quantification, enabling more accurate protein turnover determination in non-dividing cells by dynamic SILAC-based proteomics. This approach allows exact determination of protein half-lives ranging from 10 to >1000 h. We identified 4000–6000 proteins in several non-dividing cell types, corresponding to 9699 unique protein identifications over the entire data set. We observed similar protein half-lives in B-cells, natural killer cells and monocytes, whereas hepatocytes and mouse embryonic neurons show substantial differences. Our data set extends and statistically validates the previous observation that subunits of protein complexes tend to have coherent turnover. Moreover, analysis of different proteasome and nuclear pore complex assemblies suggests that their turnover rate is architecture dependent. These results illustrate that our approach allows investigating protein turnover and its implications in various cell types.

[1] Cellzome GmbH GlaxoSmithKline Meyerhofstraße 1, 69117 Heidelberg, Germany. [2] Structural and Computational Biology Unit European Molecular Biology Laboratory Meyerhofstraße 1, 69117 Heidelberg, Germany. [3] Genome Biology Unit European Molecular Biology Laboratory Meyerhofstraße 1, 69117 Heidelberg, Germany. Toby Mathieson, Holger Franken, Jan Kosinski and Nils Kurzawa contributed equally to this work. Correspondence and requests for materials should be addressed to M.Beck (email: martin.beck@embl.de) or to M.Bantscheff (email: marcus.x.bantscheff@gsk.com) or to M.M.S. (email: mikhail.savitski@embl.de)

Recent years have seen unprecedented progress in mass spectrometry-based proteomics[1]. This has enabled development of various new methodologies for interrogating the proteome. These include assessment of relative protein expression[2], detection of protein ligand interactions[3,4], monitoring changes in the abundance of post-translational modifications[5], protein half-life determinations[6–9], and many others.

In order to continue improving proteome-wide characterization of proteostasis[6,7,10,11], a further development of experimental and computational[12,13] quantitative mass spectrometry[14] work flows is required. For instance, when using dynamic SILAC (stable isotope labeling by amino acids in cell culture) to measure global protein turnover[6,15], precise and accurate peptide ion intensity quantification is needed, since even small deviations in the accuracy of measured fold changes can have a pronounced effect on the half-life measurement. In particular, when measuring protein turnover in non-dividing cells[16], many proteins will exhibit very-slow turnover because the continuous replication of the entire proteome, which occurs in exponentially growing cells is not required. As primary cells can only be kept in culture for a limited amount of time before adapting to the cell culture conditions or going into senescence, protein turnover determinations have to be based on relatively short-term treatments with stable isotope-encoded amino acids. Consequently, accurate and precise quantification is required in order to allow accurate determination of protein half-lives. We, therefore, developed procedures based on a better utilization of the isotopic distributions of ionized peptides to improve the accuracy and precision of peptide ion intensity-based quantification.

We applied this peptide ion intensity quantification strategy to analyze mass spectrometry data from dynamic SILAC experiments[17] performed in five different, non-dividing cell types: B-cells, monocytes, natural killer (NK) cells, hepatocytes, and mouse embryonic neurons to calculate protein half-lives as previously described[6]. We used this data set to validate and extend the previous observation[18] of coherent subunit turnover of protein complexes, but also observed complex architecture-dependent protein half-life distributions.

To demonstrate the usefulness of our data as a resource, we examined some exemplifying protein complexes in more detail. In agreement with previous literature[19,20], we found that histone proteins, aside from some notable exceptions in hepatocytes, have extremely slow turnover. Both, proteasomes and nuclear pore complexes (NPCs), show a clear subcomplex-dependent turnover of their subunits. The extreme longevity of the NPC previously reported in vivo for brain tissue[16], is not observed for any of the cell types investigated in vitro in this study. These results emphasize that slow NPC turnover is not a general phenomenon occurring in all non-dividing cells, but that specific NPC turnover mechanisms might exist. We conclude that our data set is a useful resource for the scientific community and our method can be broadly applied in the future.

## Results

### Improvement of peptide ion-based protein quantification.
Protein half-life determination in non-dividing cells requires precise and accurate measurement of protein fold changes. In non-dividing cells the incorporation of heavy isotope labels will be very slow for some proteins, resulting in very-low new-to-old protein ratios because only a very-small portion of the isotope has yet been incorporated. As a consequence, the ratio determination is error prone, particularly at the early time points. Such data might be stringently filtered to select for high-confidence measurements, but at the cost of coverage, specifically affecting long-lived proteins. To achieve accurate protein half-life

measurements with good coverage for long-lived proteins in primary cell systems, we investigated and optimized the parameters, which are relevant for determining very reproducible and accurate protein fold changes for the greatest possible number of proteins.

We introduced two innovations into the data analysis workflow. First, we use the exact elemental composition of the identified peptides for calculating the theoretical isotopic envelope (see methods). We established that this method yields a more accurate representation of peptide ion intensities as compared to the previously used averagine model[21] (Supplementary Figure 1) that relies on a scaled version of the average amino acid (Averagine), but does not take into account that the composition of an individual peptide might well differ from the average. Second, we used a measure to detect overlapping isotopic distributions of different peptides[22,23] (see methods), to improve the quality of quantification (Supplementary Figures 1–3). Comparison of this quantification strategy with the widely used MaxQuant software shows that our methodology performs significantly better for cases when very-low protein ratios need to be quantified, and more accurately determines the protein fold changes (Supplementary Figure 4). For proteins where the fold changes are less pronounced both softwares perform equally well. This quantification strategy is, thus, well suited for determining protein half-lives of long-lived proteins in non-dividing cells.

### Protein half-life determination.
Five different non-dividing cell types, human B-cells, monocytes, NK cells, hepatocytes, as well as mouse embryonic neurons were each labeled with medium containing $^{13}C_6^{15}N_4$ K and $^{13}C_6^{15}N_4$ R (heavy SILAC) for four different time intervals ranging between 6 and 72 h (Fig. 1; Supplementary Data 1). Taken together across all five cell types, protein half-lives were determined for a total of 9699 unique protein groups (Supplementary Data 2). The coverage in individual cell types ranged from 4667 protein groups identified in NK cells to 6534 protein groups identified in mouse neurons. To specifically query and cross-compare the high-quality subset of the data, we selected protein half-lives where the regression line (the rate constant of the protein degradation) fitted to fold changes at different time points had a Pearson correlation coefficient $R^2$ of >0.85[6,9]. This criterion was fulfilled by 8804 proteins taken together across the five cell types. The mean half-life was calculated for proteins present in both biological replicates for each cell type. The comparison of protein half-lives between biological replicates (Fig. 2) reveals excellent reproducibility with many proteins showing half-lives of greater than 500 h. The $R^2$ of the $\log_{10}$-transformed half-lives between replicates was 0.94, 0.92, 0.91, 0.93, and 0.93 for B-cells, monocytes, NK cells, hepatocytes, and mouse embryonic neurons, respectively. For 98% of all protein half-lives from the high-quality data set, the replicates differed by less than two-fold. The acquired data set, thus, represents the most extensive, high-quality catalog of protein half-lives in primary cells.

The protein turnover in NK cells was the slowest, with 210 proteins having high quality ($R^2$ of >0.85) half-lives longer than 500 h. The other cell types (monocytes, hepatocytes, B-cells, and mouse embryonic neurons) had, respectively, only 7, 17, 15, and 4 such proteins. Despite having an overall slower turnover rate, the relative $\log_{10}$ transformed half-lives between NK cells and B-cells, as well as NK cells and monocytes were in good agreement: $R^2$ of 0.65 and 0.63, respectively. The same holds true for monocytes and B-cells: $R^2$ of 0.56. In contrast, hepatocytes, which are not of hematopoietic lineage, showed the weakest correlation among the human cells, $R^2$ of 0.36, 0,41, and 0.36

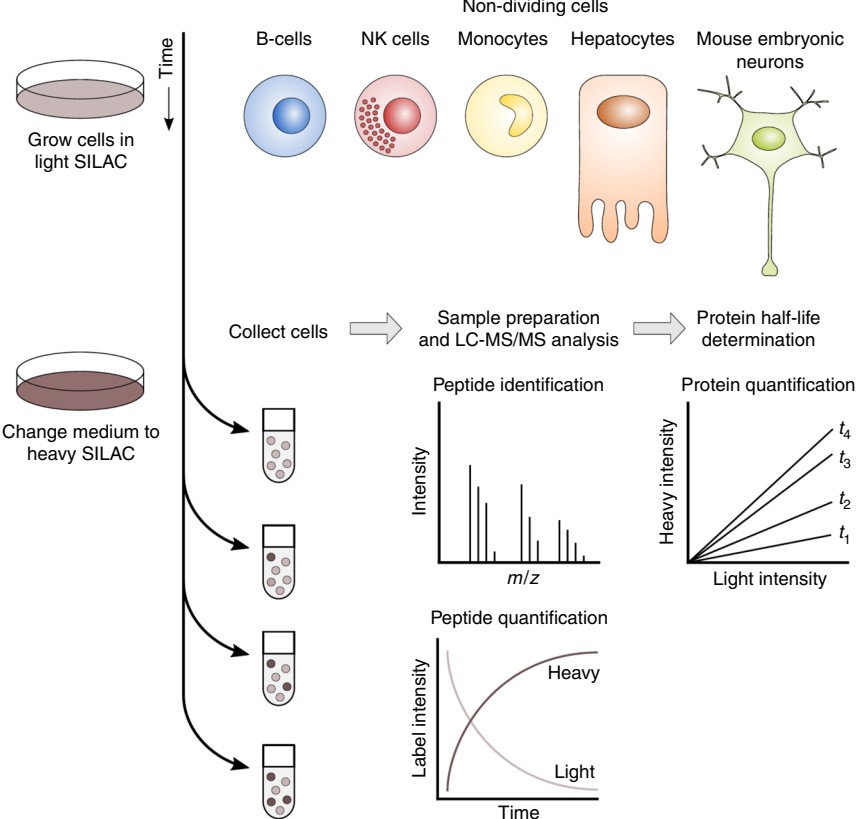

**Fig. 1** Experimental workflow for protein half-life determination using dynamic SILAC and quantitative mass spectrometry. Five non-dividing primary cell types comprising B-cells, NK cells, monocytes, hepatocytes, and mouse embryonic neurons were adapted to light SILAC medium. To label newly synthesized proteins, the cells were exposed to heavy SILAC medium and collected at different time points. After protein extraction, proteolysis with trypsin, sample preparation, and subsequent LC-MS/MS analysis, the peptides were identified by the Mascot database search engine and quantified using the isobarQuant software package. Peptides of pre-existing and newly synthesized proteins were distinguished by their mass due to incorporation of light or heavy arginine and lysine. Protein fold changes at different time points were calculated using the intensity ratios of heavy vs. light SILAC peptides and were used for subsequent protein half-life determination

when compared to B-cells, NK cells, and monocytes, respectively. Half-lives determined in the mouse embryonic neurons agreed slightly better with B-cells, NK cells, and monocytes than with hepatocytes ($R^2$ of 0.422, 0.567, 0.413 compared to 0.398). We conclude that a general categorization of the proteome in terms of protein half-lives is to quite some extent preserved over all cell types investigated here.

All protein half-lives determined in this study are summarized in Supplementary Data 2. Amongst the fast turnover proteins, we find members of the Janus family of kinases. In particular Janus kinase 3, which is predominantly expressed in the cells of hematopoietic lineage has a very short half-life between 9 and 11 h (Supplementary Figure 5) in B-cells. Among the longest-lived proteins that were reproducibly observed in more than one cell type, we find the two histone family proteins: HIST1H1C and H2AFY. The average half-life in B-cells is 2242 and 971 h, respectively (Fig. 2b, c). This value goes up to 2741 and 1950 h, respectively, in NK cells. Interestingly, these histone proteins showed a relatively very-fast turnover in hepatocytes with half-lives of 18 and 61 h, respectively. Lamin-B1 has an average half-life in B-cells of 1552 h (Fig. 2d) and in NK cells of 3215 h, the half-life in hepatocytes was faster, 388 h. In mouse embryonic neurons histone HIST1H1B has the slowest turnover, with an average half-life of 1736 h, or 72 days. Our data is in good agreement with a previous study that used radioactive labeling of the long-lived cerebral histone fractions in mice and reported half-lives of 50–100 days[24].

We also assessed the turnover behavior within the different cellular compartments, (Supplementary Figure 6). Within each compartment a broad range of protein half-lives was observed and there is a large overlap in turnover behavior between all compartments. However, significant differences in general turnover behavior are observed between the different compartments, although the effect size is small, (Supplementary Figure 6). The most prominent and significant trend is the slower turnover of the mitochondrial proteins, which is present in all cell types. Proteins from the Golgi apparatus and nucleus had reproducibly the fastest turnover, while endoplasmic reticulum and cytoplasmic proteins were located in the middle with close to identical turnover distributions.

In summary, we observe a wide spread turnover behavior within the different cell types, with particularly long half-lives observed for histones. We also observe small, but significant and reproducible differences for protein turnover in different cellular compartments.

**Turnover of protein complexes**. Our extensive data set enabled us to assess the turnover of protein complexes on a proteome-wide scale. In an earlier protein turnover study spanning 2500 proteins using $^{15}$N-pulsed labeling in mice and mass spectrometry, Price et al[18] observed a coherent turnover behavior of several protein complexes in mouse liver and brain. The complex with the best protein coverage was the 20S proteasome core

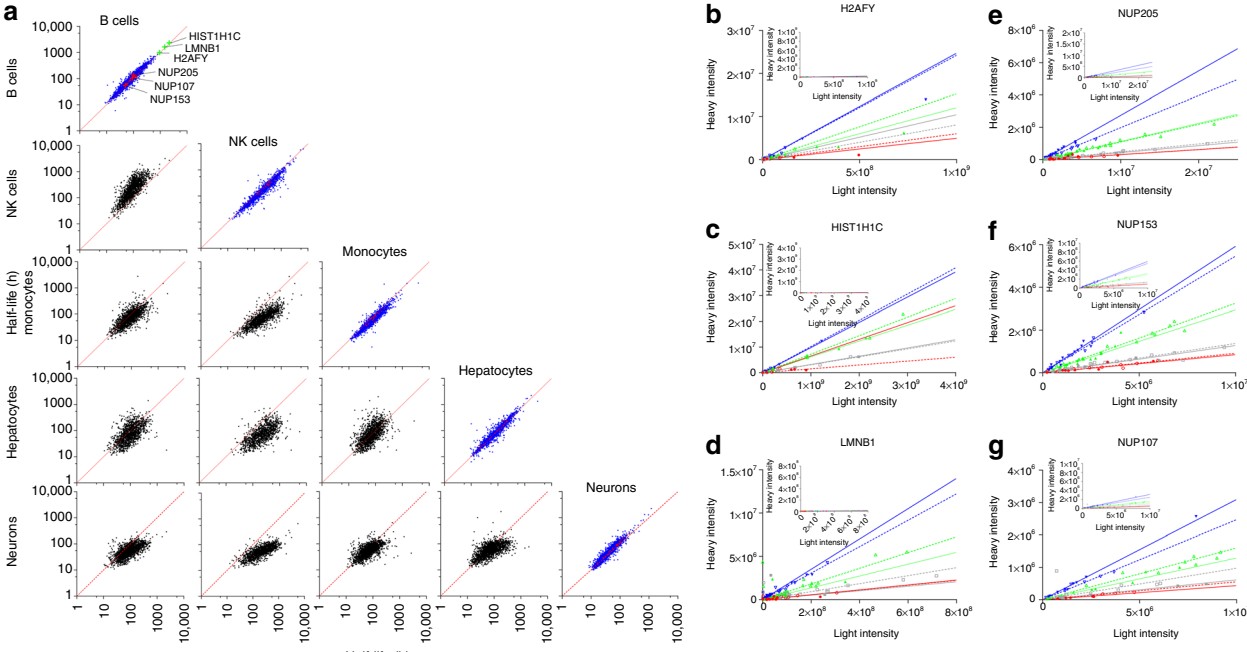

**Fig. 2** Assessment of protein half-lives in different cell types. **a** Scatter plots comparing protein half-lives determined in two experiments. On the diagonal the half-lives determined in biological replicate experiments in the same cell type are compared. Red dots indicate nuclear pore complex components. The plots below the diagonal compare the average protein half-lives determined in one cell type against other cell types as indicated. **b, c, d** Highly reproducible individual peptide fold changes from proteins with very-long half-lives determined from the B-cell experiment. The median protein fold changes are denoted by the slope of the line. The fold changes at the different time points are denoted by the different colors (red, 7 h; gray 11 h; green 24 h, and blue 34 h) with replicate one represented by a solid line and replicate two by a dashed line. The insets on each graph show the heavy and light SILAC intensities plotted on the same scale. **b** O75367 (Core histone macro-H2A.1, H2AFY) half-life from replicate one/two 945.9/995.9 h. **c** P16403 (Histone H1.2, gene HIST1H1C) half-life from replicate one/two: 2168.8/2315.5 h. **d** P20700 (Lamin-B1, gene LMNB1) half-life from replicate one/two: 1479.9/1623.5 h. The same is shown for three nuclear pore complex members (NUPs) as for (**b**, **c**, **d**). **e** Protein NUP205 has a half-life of 103.3 h in replicate one and 138.9 h in replicate two. **f** Protein NUP153 (P49790) has the shortest half-life of all the NUPs identified in B-cells with a half-life of 49.8 h in replicate one and 55.1 in replicate two. **g** Protein NUP107 (P57740) has a half-life of 91.2 h in replicate one and 104.5 h in replicate two

complex, for which a half-life of 192 h was determined in the mouse brain. In our mouse embryonic neuron data, the median half-life of the standard non-exchangeable 20S proteasome core complex subunits is 111 h, less than two-fold different compared to the value from Price et al. We performed a statistical analysis to evaluate if the observation of coherent protein complex turnover holds true when considering all complexes from an in-depth proteome turnover analysis across different non-dividing cell types. We calculated the standard deviations of the half-life values of proteins that are subunits of the same annotated complex, and compared those to the standard deviations obtained for the same complexes after the subunits were reshuffled across all complexes preserving the number of proteins in each complex group (see methods). A clear and significant trend (p-value < 0.001, Wilcoxon-rank test) for a more coherent half-life distribution of protein subunits within individual complexes becomes apparent for all cell types (Fig. 3). The chaperonin complex has the most tightly controlled turnover of the individual subunits in all different cell types. Looking at two larger complexes with more intricate architecture, the NPC and the 26S proteasome—we observe a much less tightly controlled turnover, except for the 26S proteasome in mouse neurons, where the turnover behavior is very coherent.

Looking in more detail at the 26S proteasome (Fig. 4a; Supplementary Data 3), we observe significantly different half-lives of the 20S core complex subunits compared to 19S regulatory complex subunits in all cell types, except for mouse embryonic neurons (Fig. 4a), which explains the more coherent turnover behavior in these cells. Interestingly, there is a

significant trend for the 20S core complex to be more stable than the 19S regulatory complex in B-cells, monocytes, and NK cells, but a clear and significant opposite behavior is observed for hepatocytes. Clustering of the proteasome subunits according to the similarity between their half-lives across the human cells also leads to distinct separation between the core and regulatory subunits (Fig. 4b).

Next, we looked at the NPC, one of the largest protein complexes in the cell. This complex has been studied in rat brain by Toyama[19] et al, also using an in vivo [15]N labeling strategy and nucleoporins were shown to be particularly long-lived in rat brain tissue[19]. The authors also observed that the half-life of nucleoporins correlates with their allocation in subcomplexes within the NPC. Specifically, while about 40–50% of the inner ring complex members Nup205, Nup155, and Nup93 had not turned over after 6 months, only about 10–20% of the original pool of the Y-complex members Nup160, Nup107, Nup96, and Nup85 remained detectable. To the best of our knowledge, the data set by Toyama et al. thus far accounts for the only investigation in non-dividing cells (see discussion).

In the non-dividing cells in our study, we observe a relatively quick turnover of Nups that is almost one order of magnitude faster as compared to histones (Fig. 2e, f, g; Supplementary Data 4). Generally, the half-lives of the Nups are approximately located in the middle of the distribution of all other protein half-lives of the same cell type. Mouse embryonic neurons are an exception where Nups are turning over slightly slower than most other proteins, but still much faster than histones (Fig. 2a; Supplementary Figure 7). We do not observe pronounced

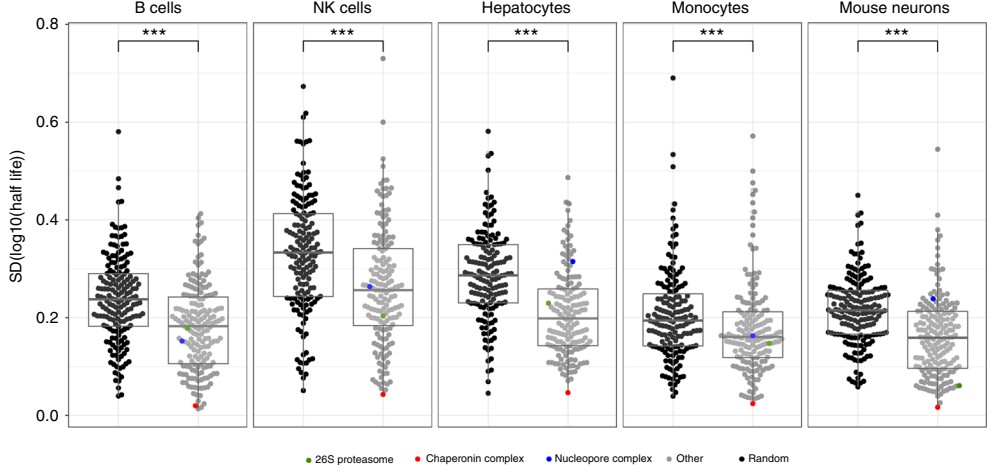

**Fig. 3** Half-life variability among members of protein complexes is smaller than expected by chance. Distributions of standard deviations (SD) of half-lives from proteins in complexes as annotated in the CORUM database compared to SD of the half-lives of the same proteins shuffled across the different complexes, while preserving the number of proteins in each complex group. Differences in the $\log_{10}$ half-lives of true protein complex members vs. the random draws of proteins in a given cell type were assessed by Wilcoxon-rank test (significance levels were encoded as *** $p < 0.001$, ** $p < 0.01$, * $p < 0.05$). Center line in box plots is the median, the bounds of the boxes are the 75 and 25% percentiles i.e., the interquartile range (IQR) and the whiskers correspond to the highest or lowest respective value or if the lowest or highest value is an outlier (greater than 1.5 * IQR from the bounds of the boxes) it is exactly 1.5 * IQR

differences between the inner ring and Y-complex members; e.g., in B-cells the majority of all members turn over just above 100 h in both subcomplexes. In our comprehensive data set, we do, however, observe a general clustering of half-lives into known subcomplexes (Fig. 5a, b). The half-lives of members of the Nup358 complex, and to some extent also the Nup214 complex, are generally shorter when compared to the inner ring and Y-complexes. The half-lives of members of the Nup62 complex, although spatially positioned in the inner ring complex, appear to be uncoupled from the latter. In hepatocytes and monocytes it is more short-lived but in B-cells more long-lived when compared to other inner ring Nups (Fig. 5a). Interestingly, the turnover of Nup188 is in line with those of the Nup214 complex and Nup98, and an association of which has been proposed[25].

Partitioning the nucleoporins into a scaffold and a peripheral group (Supplementary Data 4) and comparing the half-life distributions between the two groups shows a statistically significant trend for faster turnover of the nucleoporins in the peripheral group for all cell types (Fig. 5a). In agreement with previous work[19], we find that Nup98 turns over considerably more quickly than Nup96, although both proteins are synthesized as a single fusion protein prior to autoproteolytic cleavage. This might be explained by the existence of an additional transcript encoding only Nup98. Nup153, Nup50, and the transmembrane Nup gp210 have been shown to have short mean residence times at the NPC[26], although this of course does not mean that they turn over once they disassociate. Interestingly, both Nup153 and Nup50 have relatively short half-lives, e.g., 50–70 h in B-cells. In striking contrast, gp210 generally lives at least as long as scaffold Nups, e.g., ~230 h in B-cells.

## Discussion

In order to accurately quantify protein turnover in non-dividing cells, we introduced and validated new concepts for peptide ion intensity-based quantification. The methodological improvements are made publicly available as part of the isobarQuant package[27] (https://github.com/protcode/isob/archive/1.1.0.zip)), which supports all existing approaches for peptide ion intensity-based quantification, such as SILAC[28], mTRAQ[29], and dimethyl

labeling[30,31] and allows for adaptation to any type of peptide ion-based quantification strategy. In total we determine half-lives for 9699 proteins taken together from five different non-dividing cell types; B-cells, monocytes, NK cells, hepatocytes, and mouse neurons and make these publicly available as a resource to the scientific community.

In our extensive study of protein turnover in non-dividing cells, we observed very-long half-lives for histone proteins in accordance with previous in vivo work in non-dividing brain cells[16] on a relative scale. Similarly, the turnover of proteasomes observed in vivo[18], is well in line with our data. For NPCs, Toyama et al. observed an extreme longevity similar to histones in rat brain in vivo[16]. This is not true for any of the non-dividing cell types investigated in in vitro in this study, but relatively short half-lives of Nups, both on an absolute scale and relative to histones are observed. The technical details and also the biological context of both experiments are very difficult to compare, and either of them have its own benefits. The half-lives measured in non-dividing cells in vitro should yield more accurate values at the cost of losing the endogenous in vivo context, which, of course, is also important. We can nevertheless conclude that the very-slow NPC turnover is not a general phenomenon of all non-dividing cells. This might have important biological implications. Although an NPC surveillance pathway for defective NPC assembly intermediates has been described for yeast[32], the turnover of fully assembled NPCs is believed to be extremely rare[33] and a corresponding pathway has, to the best of our knowledge, not yet been identified. One might thus speculate that NPC-specific turnover or maintenance pathways might exist that remain to be characterized. It will thus be interesting to look out for NPC turnover in non-dividing cells in the future. One would predict that both should occasionally occur.

For both the NPC and also the proteasome, we observe protein complex architecture-dependent turnover. The 19S regulatory and the 20S core particle have entirely independent assembly pathways[34], which would explain the uncoupled turnover, but also indicate that cap and core associate dynamically. An interesting pattern is observed for the beta-type subunits of the 20S core particle that are specifically integrated into immuno-proteasomes[34] and show a different turnover in several cell types

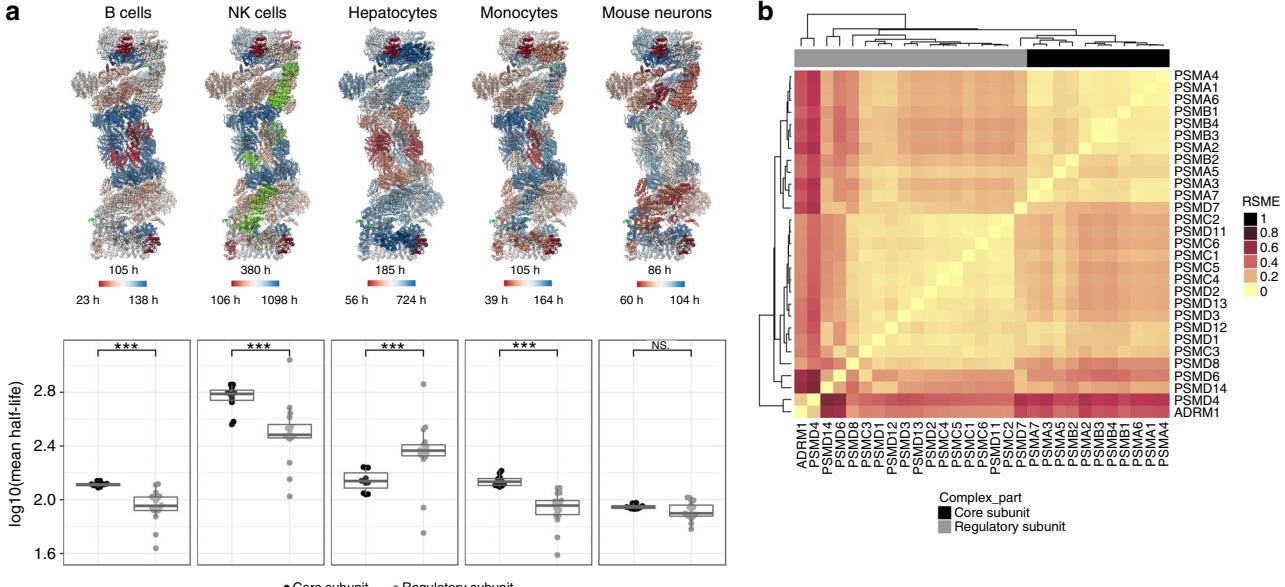

**Fig. 4** Architecture-dependent turnover of the proteasome subunits. **a** Upper panel: proteasome subunits are shown color-coded as gradient from red (short half-life) to blue (long half-life). For each cell type, half-lives were averaged over biological replicates, except for rare cases where only one half-life value was available, and converted to a color gradient as explained in the Methods. The median, minimum, and maximum half-lives are indicated together with the color bars. Subunits with undetermined half-lives are colored green. Lower panel: distributions of the reproducibly measured half-lives of the regulatory and the non-exchangeable core subunits of the proteasome in the different cell types. Differences in the distributions of $\log_{10}$ half-lives were assessed by Wilcoxon-rank test (significance levels were encoded as *** $p < 0.001$, ** $p < 0.01$, * $p < 0.05$). Center line in box plots is the median, the bounds of the boxes are the 75 and 25% percentiles i.e., the interquartile range (IQR) and the whiskers correspond to the highest or lowest respective value or if the lowest or highest value is an outlier (greater than 1.5 * IQR from the bounds of the boxes) it is exactly 1.5 * IQR **b** Heatmap showing the comparison for each pair of proteasome subunits by calculating the root mean square error between their four $\log_{10}$ transformed half-lives in the four different human cell types. Hierarchical clustering leads to separation of the regulatory subunits from the non-exchangeable core subunits. The 19S proteasome subunits PSMD4 and the recently discovered ADRM1 form a distinct cluster

investigated here (Fig. 4a). Since it is difficult to conceive that these subunits dynamically exchange with the assembled 20S core, one might interpret this as two different populations of proteasomes that share most, but not all of their subunits and have different turnover times.

Our observations both for the proteasome and the NPC suggest that there is a trend for shorter half-lives of peripheral subunits. This is exemplified by the 19S proteasome subunits PSMD4 and ADRM1[35] that are clear outliers. They turn over much quicker than the rest of the 19S subunits and have nearly identical variation in their half-life patterns across the four human cell types (Fig. 4b; Supplementary Data 3). These two ubiquitin receptor proteins are located in the distal part of the regulatory particle, and are postulated to have been recruited to the complex late in its evolution[35]. Also, in the case of NPC, three of the β-propeller proteins of the Y-complex, Nup37, Nup43, and Elys, are not always correlated with the remaining subcomplex members (Fig. 5a). This is most apparent in hepatocytes, where they are much more short-lived. Interestingly, these three proteins are not conserved across the tree of life, and, thus, might be regarded as more auxiliary additions to the essential core of the Y-complex. A possible interpretation is that the peripheral complex subunits that have joined later in complex evolution are less strongly associated with the complex and more easily replaced.

## Methods

**Primary cell isolation and treatment.** Primary human hepatocytes purchased from KaLy-Cell, kept in Williams E medium (ThermoFisher, 22551022) supplemented with 10% dialyzed FBS and with primary hepatocyte maintenance supplement (ThermoFisher, CM4000), as well as human monocytes, B-cells, and NK cells, isolated from peripheral blood mononuclear cells, derived from buffy coats (German Red Cross, Mannheim) by magnetic-bead based negative selection (STEMCELL Technologies), and resuspended in RPMI-1640 medium (Thermo-Fisher, 21875-034) supplemented with 10% dialyzed FBS, were cultured in the presence of light (L) SILAC amino acids lysine and arginine overnight at 37 °C. The cells were then pulse-labeled with corresponding media containing heavy (H) isotope-labeled amino acids (lysine, ($^{13}C_6{}^{15}N_2$, Sigma-Aldrich, 608041) and argi-nine ($^{13}C_6{}^{15}N_4$, ThermoFisher, 88434)) for the indicated time periods (Supple-mentary Data 1), washed, pelleted, and snap-frozen in liquid $N_2$. The cell pellets were lysed in buffer containing 4% SDS and digested with benzonase.

**Primary neuron culture.** Cortical neuronal cells were isolated from prenatal embryos of CD-1 mouse at embryonic day 15 (E15). To dissociate the cortex tissue, it was finely chopped by scalpel followed by digestion in Accutase (ThermoFisher, A1110501) for 12 mins. To prevent clumping due to DNA from dead cells, the tissue was treated with 250 unit/μl of Benzonase (Millipore, 71206-3). Neurons were triturated gently with a fire-polished Pasteur pipette and passed through the 40 μm cell strainer (BD Falcon, 352340) before plating them onto 6-well plate at a density of $1 \times 10^6$ cells per well. The plates were coated with 0.1 mg/ml of Poly-D-Lysine (Sigma, P0899) and 2.5 μg/ml of laminin (Sigma, 11243217001). Cultures were maintained in Neurobasal medium (ThermoFisher, 21103) containing 1% penicillin/streptomycin (ThermoFisher, 15140122), 1% GlutaMAX (ThermoFisher, 35050), and 2% B27 supplement (ThermoFisher, 12587) at 37 °C with 5% carbon dioxide in the incubator. Post-seeding after 1 day in vitro (DIV 1), half of the medium was replaced with fresh pre-warmed Neurobasal medium with all the supplements (above). On DIV 4, the neurons were treated with 1.0 μM of cytosine arabinoside (Tocris, 4520/50). On DIV 5, dynamic SILAC experiments were started by exchanging one-fifth of the medium with a final 10 × excess of heavy lysine ($^{13}C_6{}^{15}N_2$, Sigma-Aldrich, 608041) and heavy arginine ($^{13}C_6{}^{15}N_4$, ThermoFisher, 88434). The cells were harvested 0, 6, 12, 24, or 36 h after pulse, washed with PBS including protease inhibitors (Sigma-Aldrich, CO-RO Roche), and lysed in 50 mM TRIS-HCl, pH7.4, supplemented with 4% SDS and benzonase. Lysates were cleared by centrifugation at 20.000x $g$ at room temperature, followed by protein concentration measurement (BCA assay, ThermoFisher, 23225). A volume of 20 μg protein of each time point were used for MS analysis.

All animal experiments were conducted under veterinarian supervision and rules of the European Molecular Biology Laboratory, following the guidelines of the European Commission, revised directive 2010/63/EU and AVMA Guidelines 2007.

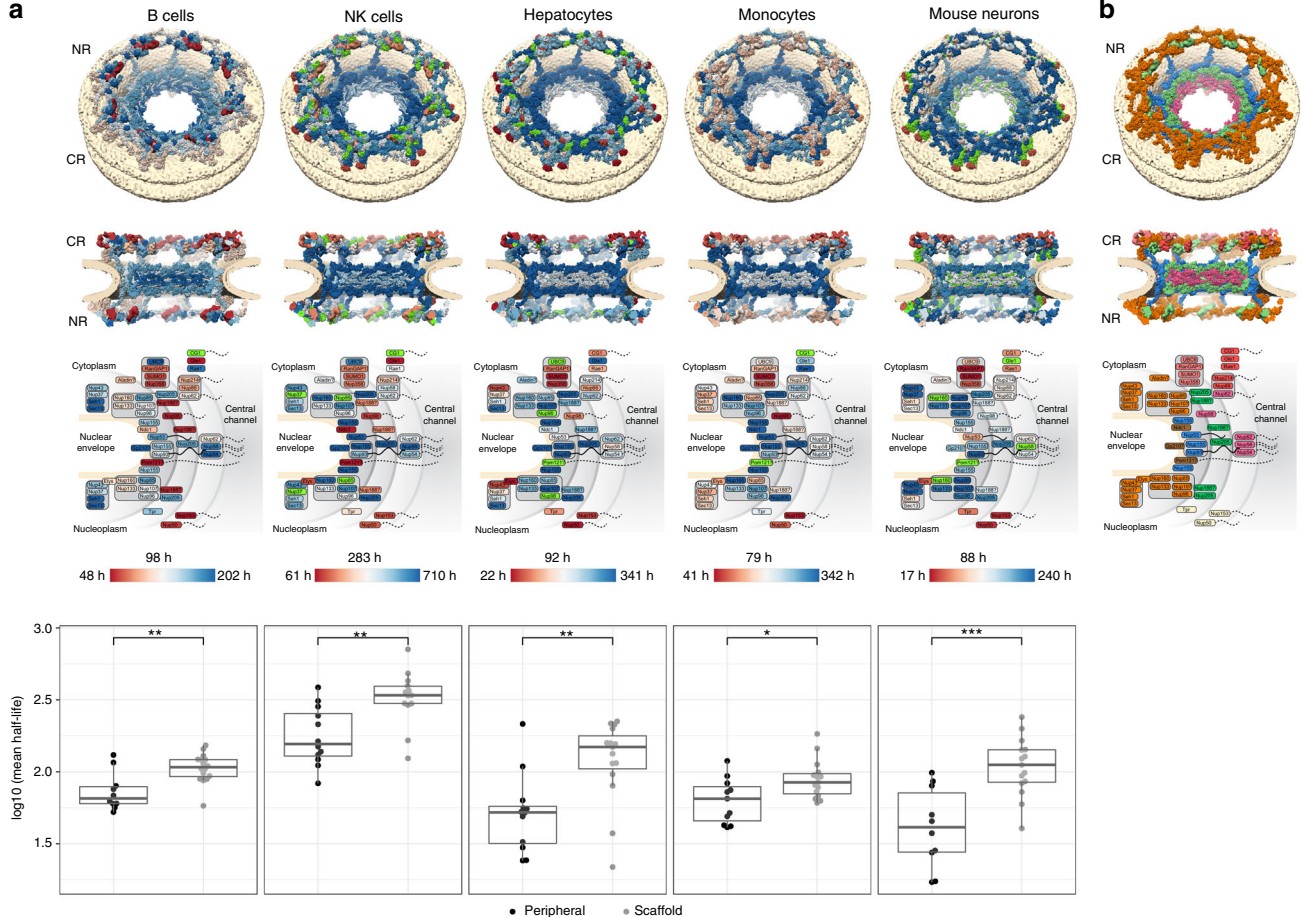

**Fig. 5** Architecture-dependent turnover of the nuclear pore subunits. **a** Upper panel: nucleoporin half-lives mapped onto the structure of the nuclear pore complex. Nups are shown color-coded as gradient from red (short half-life) to blue (long life-life). An architectural model of the nuclear pore[42, 43] is shown as seen from top (top panel), cut in half (middle panel), and a subcomplex scheme (bottom panel). The nucleoplasmic side is at the bottom in all cases. For each cell type, half-lives were averaged over two biological replicates, except for rare cases where only one half-life value was available, and converted to a color gradient as explained in the Methods. The median and minimum and maximum half-lives are indicated together with the color bars. Nups with non-determined half-lives are colored green. Lower panel: distributions of the reproducibly measured half-lives of the scaffold and peripheral subunits of the nuclear pore in the different cell types. Differences in the distributions of $\log_{10}$ half-lives were assessed by Wilcoxon-rank test (significance levels were encoded as *** $p < 0.001$, ** $p < 0.01$, * $p < 0.05$). Center line in box plots is the median, the bounds of the boxes are the 75 and 25% percentiles i.e., the interquartile range (IQR), and the whiskers correspond to the highest or lowest respective value or if the lowest or highest value is an outlier (greater than 1.5 * IQR from the bounds of the boxes) it is exactly 1.5 * IQR **b** The same as upper panel in **(a)**, but color-coded according to Nup subcomplexes. Nups of the inner ring are colored blue, of the outer (Y-complex) rings—orange, trans-membrane nucleoporins—brown, Nup205 and Nup188—green, nuclear basket nucleoporins—yellow, Nup62 subcomplex—magenta, Nup358 subcomplex—salmon, and Nup214 complex—red

**THP-1 SILAC cell mixtures for method evaluation**. THP-1 cell cultures (ATCC TIB-202) were established in RPMI-based SILAC media and supplemented with either light or heavy isotope-labeled amino acids (as above). For harvesting, the cells were washed, pelleted, and snap-frozen in liquid $N_2$. They were lysed with 4% SDS in 50 mM Tris-HCl pH 7.4 and the DNA was digested with benzonase nuclease (Sigma, E1014-25KU). Three independent dilution series (1:1, 1:9, and 1:49) were created by mixing the 25, 5, and 1 µL, respectively, of SILAC-H to SILAC-L medium to a final volume of 50 µL.

The cell line was checked for mycoplasma contamination and authenticated using the Promega kit MycoAlert.

**Sample preparation for MS**. Cells were washed with PBS and the supernatant was removed completely before cells were lysed in 2% SDS for 3 min at 95 °C in a thermomixer (Thermo Fisher Scientific), followed by digestion of DNA with Benzonase at 37 °C for 1.5 h. Lysate was cleared by centrifugation and the protein concentration in the supernatant was determined by BCA assay. Proteins were reduced by DTT and alkylated with iodacetamide, separated on 4−12% NuPAGE (Invitrogen), and stained with colloidal Coomassie[36] before proceeding to trypsin digestion and mass spectrometric analysis (see below). Gel lanes were cut into three slices covering the entire separation range (~2 cm) and subjected to in-gel digestion[4]. Peptide extracts were additionally fractionated on an Ultimate3000 (Dionex,

Sunnyvale, CA) using reversed-phase chromatography at, pH 12, 1 mm Xbridge column (Waters, Milford, MA), as previously described[37].

**LC-MS/MS analysis**. Samples were dried in vacuo and resuspended in 0.05 % trifluoroacetic acid (TFA) in water. Of the sample, 50% was injected into an Ultimate3000 nanoRLSC (Dionex, Sunnyvale, CA) coupled to a Q-Exactive plus (Thermo Fisher Scientific). Peptides were trapped on a 5 mm × 300 µm C18 column (Pepmap100, 5 µm, 300 Å, Thermo Fisher Scientific) in water with 0.05% TFA at 60 °C. Separation was performed on custom 50 cm × 100 µM (ID) reversed-phase columns (Reprosil) at 55 °C. Gradient elution was performed from 2% acetonitrile to 40% acetonitrile in 0.1% formic acid and 3.5 % DMSO over 2 h. The samples were online injected into Q-Exactive plus mass spectrometers operating with a data-dependent top 10 method. MS spectra were acquired using 70,000 resolution and an ion target of $3 \times 10^6$. Higher energy collisional dissociation (HCD) scans were performed with 25% NCE at 17,500 resolution (at $m/z$ 200), and the ion target setting was fixed at $1 \times 10^6$. The instruments were operated with Tune 2.3 and Xcalibur 3.0.63

**Peptide precursor intensity-based quantification**. All acquired, raw data were processed using a modified-version of isobarQuant[27] available from the Github code repository https://github.com/protcode/isob/archive/1.1.0.zip: further

information about which can be found in the user manual (see supplementary information). The configuration (cfg) files were set up for SILAC and processed as described hereafter. The QuantMethod.cfg file was provided with a new quantification method for SILAC (silac3). This method contains the mass information for LIGHT (K + 0, R + 0), MEDIUM (K + $^{13}C_6$, R + $^{13}C_6$), and HEAVY (K + $^{13}C_6$ + $^{15}N_2$, R + $^{13}C_6$ + $^{15}N_4$) SILAC modifications on lysine and arginine. The quantification source was set to MS1 and the number of threads in which to run the software was set according to the number of processors on the workstation. This method preset is supplied with the isobarQuant package.

**Pre-Mascot workflow.** In the pre-Mascot step of isobarQuant, all acquired MS1 and MS2 information along with instrument and HPLC runtime information and acquisition parameters from the Xcalibur raw files were transferred to a file in HDF5-format (http://www.hdfgroup.org). Ion chromatograms of each MS2 event are recorded in the HDF5 file and later used to recalculate an accurate mass of the precursor given by Xcalibur by selecting the mass at the highest intensity of the monoisotopic peak. The reassigned parent masses were written to a file in Mascot Generic File (mgf) format along with the relevant deisotoped and deconvoluted MS2 spectra[38,39] and searched with Mascot 2.5 via the Mascot Daemon version 2.5.1 (Matrix Science, Boston, MA) against the October 2014 release of Human Uniprot Proteome appended with a decoy version of the same using a 10 ppm (parts per million) mass tolerance for peptide precursors and a 20 mDa (milliDalton) mass tolerance for fragment ions (since high-resolution data were acquired in HCD mode in the Orbitrap). Carbamidomethylation of cysteine residues was selected as a fixed modification, and the following were selected as variable modifications: oxidation of methionine, acetylation of protein N-termini, SILAC heavy label 13C(6) 15N(4) on arginine (+10.008269 Da) and SILAC heavy label 13C(6) 15 N(2) on lysine (+8.014199 Da). Upon completion of the Mascot search, an auxiliary Python script, available as part of the isobarQuant package, was called via the Actions/External Processes option within the Mascot Daemon to transfer the result (dat) file from the Mascot server to a user-supplied directory, prior to processing using the post-Mascot workflow of isobarQuant.

**Quantification of peptides and calculation of intensity fits.** The post-Mascot workflow of isobarQuant was started in 'mergeresults' mode to merge data acquired from the multiple offline fractionation steps. The first part of this workflow extracted the relevant data from the dat files and matched it to the acquired raw data via the Xcalibur scan ID. Following this internalization, the precursor intensity (MS1)-based quantification proceeded as follows:

1. Based on the peptide sequence, charge state and any modifications not related to quantification (e.g., heavy SILAC arginine) all Mascot-identified SSMs from that search of that raw file are condensed into groups, with the highest Mascot scoring SSM representing each group as a single record. This single record is referred to as a PCM (for Peptide, Charge state, Modification string)
2. The PCMs were processed in order of ascending retention time (RT). The theoretical isotope masses were calculated for each label state. The raw extracted ion chromatograms (XICs) present within a one minute window around the PCM's retention time were extracted from the raw data for each isotope mass.
3. Chromatographic peaks in each XIC were detected and grouped to form isotopic clusters for each label state. The grouping involved the mapping of overlapping peaks identified from each isotope XIC to peaks identified from the monoisotopic XIC. The peaks are only considered to be overlapping when the RT spread between the two 50% apex intensity points of the peaks overlap. All possible clusters are generated before removing any where the first $^{13}C$ isotope is missing. Isotope clusters are also generated from the preceding survey spectrum (PS) just prior to the MS2 spectrum acquisition (of the PCM) and also from the survey scan at the apex (AS).
4. According to the given PCM, an exact model of the intensities of the theoretical isotopic envelope, based on its elemental composition, was constructed. This is depicted in Supplementary Figure 1A and is carried out for all available labels (here: the heavy and light SILAC versions of the PCM).
5. A least squares method was used to find the isotopic cluster with the best fit to the exact model from the list of candidates in the XIC data. This yields two values: a fitted intensity used for quantification and a measure of the quality of the fit, calculated as the sum of the squares of the residual values. If the fit of the best of the XIC clusters is greater than 0.1 it is compared to the fits obtained from the AS and PS and the best fitted result of the three is selected.
6. The quantification value, reported and stored in the HDF5 file, is the sum of the theoretical intensities calculated for the label multiplied by the fitted intensity.

For ease of understanding, the PCMs used above are referred to simply as peptides in the rest of this manuscript.

**Calculation of prior ion ratio.** We define the 'prior ion' as the peak occurring at an $m/z$ corresponding to the loss of one neutron from the monoisotopic ion. This peak is not expected to be present in light peptides and to have a low intensity for heavy peptides because it is usually the result of incomplete incorporation of heavy atoms

into the heavy SILAC label; this is shown in left panel of Supplementary Figure 1B. If an intense peak is present at this position it indicates that a co-eluting (interfering) isotope cluster is present and that the fitted intensity is likely to be overestimated (right panel of Supplementary Figure 1B). We use the term 'prior ion ratio' to describe the ratio between the 'prior ion' intensity and the sum of all fitted intensities in the corresponding isotopic cluster. This value was calculated for all fits determined in the previous step and its influence on precision and accuracy was investigated during this study.

**Protein inference.** Protein groups were inferred from a high-quality subset of the peptide identifications made by Mascot. All of the peptides must pass the 1% (peptide) false discovery rates (FDR) criterion. In addition, at least one peptide of minimum length seven with a Mascot score difference to the next highest scoring non-identical peptide sequence of at least 10 points must be present in the group. Any groups consisting only of peptides present in other groups with higher total scores (Mascot light red peptides) were removed and all groups consisting of peptides fully subsumed by other protein groups were aggregated. Those protein groups sharing a Uniprot gene identifier were merged together (an option in isobarQuant). Peptides passing the FDR cutoff (1%) were termed unique if their peptide sequence was identified in one distinct protein group.

Protein-based FDR were calculated using the picked-protein FDR approach[40], which has been developed to avoid overestimating the number of false positive (decoy) hits in the data set and is incorporated into this release of isobarQuant. This simple method avoids inflating the number of decoy hits in a data set by 'picking' the protein accession corresponding to the highest score when a pair (target and decoy of the same accession) have both been identified and is automatically calculated by isobarQuant.

**Protein quantification.** Unique peptides quantified in the previous part of the workflow and associated with each protein group were selected and filtered (unless otherwise stated) according to the settings given in the corresponding configuration file (proteinquantfication.cfg): Mascot score >15, peptide length ≥6, FDR at Mascot score <1%, maximum least squares fit of both peptides in light/heavy pair ≤0.1, and maximum prior ion ratio of both peptides in light/heavy pair ≤0.2. The remaining quantification filters in isobarQuant, related to MS2-based quantification strategies, were not switched on (set to −1). Protein fold changes were calculated using the median of all valid quantified peptide ratios (the fitted intensity of heavy SILAC to light SILAC) linked to the protein group. For protein groups where there was a minimum of one peptide with a positive ratio, any peptides with an undetermined ratio were excluded from the median calculation. The parameters described here are set as the default setting in isobarQuant and have been optimized during this investigation (see Results).

Protein fold changes determined at different time points were extracted from the output of isobarQuant and used to calculate the protein half-lives.

**Comparison between isobarQuant and MaxQuant.** In order to compare quantification precision and accuracy to a widely used software, we also analyzed THP-1 SILAC mixtures with both IsobarQuant and MaxQuant. Acquired data were processed with MaxQuant (Version1.5.8.0) using the settings generally recommended for SILAC quantification[22]. Searches were performed against the same search database as above using carbamidomethylation of cysteine as a fixed modification and oxidation of methionine and acetyl (protein N terminus) as variable modifications. The mass tolerance for the precursor was 4.5 ppm and 20 ppm for the fragment ions, 're-quantify' option was switched on[22], (a separate analysis was also performed with the 're-quantify' option switched off), in the section 'group-specific parameters' a multiplicity of two was selected, with Arg10 and Lys8 chosen as the 'Heavy labels'. A complete list of all MaxQuant settings is available in (Supplementary Data 5).

**Protein half-life determination.** Protein half-lives were determined according to a modified-version of the protein decay rate method described by Schwanhäusser et al[6]. Since we were working with non-diving cells, the cell cycle time correction component was removed from the equation to give the formula:

$$k_{\mathrm{dp}} = \frac{\sum_{i=1}^{m} \log_e(r_{t_i} + 1) \cdot t_i}{\sum_{i=1}^{m} t_i^2}$$

where $k_{\mathrm{dp}}$ is the rate constant of the protein decay, $m$ is the number of time points ($t_i$) considered and $r_{ti}$ is the fold-change ratio (heavy / light) of a specific protein at each time point. The half-life of a protein ($T_{1/2}$) is then calculated by

$$T_{1/2} = \frac{\log_e 2}{k_{\mathrm{dp}}}$$

For each protein a linear model was fitted to the time course of the logarithmic protein fold changes [$\log_e(r_{ti} + 1)$] and the coefficient of determination ($R^2$) for the

linear regression was recorded (Supplementary Figure 1). The QC value was set to 'weak' if it was possible to determine a fold change in at least three out of the four time points, 'good' if the protein fold changes at three out of the four time points were based on a minimum of three quantified peptides, and 'poor' for the remainder.

**Analysis of half-life variability within protein complexes.** All proteins identified in any of the cell types were mapped to complexes using the CORUM database[41]. The complexes were filtered to contain at least five quantified protein members in a given cell type and the standard deviation of the $\log_{10}$ transformed half-lives among the complex members was computed. In order to create a reference that indicated half-life variability as expected by chance, random representatives were drawn from the list of proteins associated with the complexes in a given cell type and placed into groups representing the different numbers of true complex members. Thus, in the end, a random group of proteins associated with a complex would have the same number of proteins as the original, true complex group. As previously the standard deviation of the $\log_{10}$ transformed half-lives among these proteins within each group was computed. Differences in the $\log_{10}$ half-lives of true protein complex members vs. the random draws of proteins in a given cell type were assessed by Wilcoxon-rank test (significance levels were encoded as *** $p < 0.001$, ** $p < 0.01$, * $p < 0.05$).

**Mapping of protein half-lives on protein complex structures.** For each cell type, the mean protein half-lives calculated from two replicates were mapped onto protein complex structures as a linear three-color gradient. To avoid distorting the gradient by outliers with exceptionally high or low half-lives, the gradient midpoint was set to the median half-life of the subunits of the complex and the lower and upper half-life values for the linear interpolation were set to the 15th and 85th percentile, respectively. The half-lives outside this percentile range were clipped. The median and percentiles were calculated as a mean of, respectively, the medians and percentiles of each biological replicate. For coloring the Nup214 complex, which is represented in the structure as a single density segment composed of three subunits, the mean half-life of the three subunits was used. All calculations were performed using half-lives with an $R^2$ of at least 0.25.

**Code availability.** The isobarQuant code is freely available from the Github code repository: https://github.com/protcode/isob/archive/1.1.0.zip

**Data Availability.** All mass spectrometry data (571 raw files) and their corresponding Mascot search files have been uploaded to the Proteomics Identifications database (PRIDE) at http://www.ebi.ac.uk/pride/, and are under accession numbers PXD008511 (B-cells), PXD008512 (hepatocytes), PXD008513 (monocytes), PXD008514 (mouse neurons), PXD008515 (NK cells) and PXD008516 (THP-1 cells). All other data are available from the corresponding authors upon reasonable request.

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

## Acknowledgements

We would like to thank J. Stuhlfauth for cell culture, Katrin Strohmer, M. Jundt, K. Kammerer, M. Klös-Hudak, M. Paulmann, and T. Rudi for expert technical assistance.

## Author contributions

T.M., M.Ba. and M.M.S. conceived the project; D.P., G.B., M.Ba. and M.M.S. designed the experiments; N.Z., D.P., V.S.R., M.Sc., I.B., M.St. and K.-M.N. conducted and supervised experiments; T.M., G.S. and H.F. developed the software; T.M., H.F., J.K., N.K., M.Be., M.Ba. and M.M.S. performed data analysis; H.F., N.K. and J.K. contributed to the manuscript; and T.M., M.Be., M.Ba. and M.M.S. wrote the manuscript.
