## [Peer Review File · Nature Communications]

Reviewers' comments:

Reviewer #1 (Remarks to the Author):

Mathieson et al. describe the development of novel procedures to accurately determine protein half-lives, using a coupled SILAC and mass spectrometry approach. The authors describe substantially higher cellular turnover rates for components of the nuclear pore complex than in a previous report by Toyama et al., which employed rat brain tissue and concluded that nucleoporins are among the longest-lived proteins. Whereas most components of the NPC inner ring complex and the Y-shaped Nup84/Nup107 complex display similarly long half-lives, nucleoporins comprising the Nup358 and Nup214 complexes are rapidly turned over. The authors conclude that the varying turnover rates observed are cell type specific, suggesting different mechanisms for protein turnover within the nuclear pore complex. The reported findings are of potential interest to the readership of Nature Communications. However, the current manuscript requires substantial additional experimental validation of their new mass spec method to provide satisfactory evidence for the physiological relevance expanded upon in the conclusions. This is crucial as the implications of multiple modes of nucleoporin turnover in pre-formed pores would be of considerable interest within the field and undoubtedly will attract significant scrutiny. However, any conclusion regarding the longevity of individual nucleoporins present in otherwise long-lived complexes requires substantial evidence, which is currently insufficient, as this paper may act as the starting point towards uncovering novel machineries that regulate nuclear pore complex turnover.

Major points

(1) Large stretches of the manuscript are extremely technical and require extensive knowledge of mass spectrometry. The authors should revise the main text for increased clarity, aimed towards the broad readership of Nature Communications. In fact, a good case can be made to break up the manuscript into two parts. The authors could publish their novel approach in a specialized proteomics or mass spectrometry journal and the application of the method in a more specialized cell biology journal. As it stands the current manuscript is inaccessible to the broad readership of Nature Communications, who will be hard pressed to discern the limitations of both the methodology and results presented.

(2) The authors report dramatically different nucleoporin turnover rates compared to the findings of the Hetzer group (Toyama et al., Cell, 2013). Because of these striking differences, it is essential to validate the novel procedures that were employed in the current manuscript. The comparison with previous findings of protein turnover rates has to be expanded. To establish confidence in the new method, the authors must include analyses of alternative macromolecular complexes with established protein turnover rates.

(3) Toyama et al. previously reported that nucleoporins are particularly long-lived in rat brain tissue. The current manuscript now describes a relatively fast turnover of nucleoporins in other non-dividing cells, which is in stark contrast to the previous findings. Once the validity of the procedure is established (see above), a direct comparison to the findings of Toyama et al. is required to address this discrepancy. The reviewer is aware that Toyama et al. sacrificed rats up to 12 months post-chase, which is not a feasible timeframe for a revision. However, the authors could instead use primary rat neuronal cultures for coupled SILAC and mass spectrometry, providing a biologically relevant comparison to the previous findings. Additionally, the NPC analysis should also be comprehensive with the turnover rates of all ~35 nucleoporins presented, not just the turnover rates of a subset.

(4) In Figure 6A, the first biological replicate of the NK and Monocyte samples displays consistently higher protein turnover rates compared to the second biological replicate. The authors should

comment on the occurrence of these differences. Moreover, a third biological replicate should be considered to further strengthen the data.

(5) For several nucleoporins, turnover rates were determined in some cell types but not in others, e.g. Nup37 and Aladin. The authors should justify these differences in detail. Furthermore, the legend to Figure 7A states that protein half-lives were averaged over biological replicates. However, in Supplementary Table 3, the half-life of Nup43 was determined only in biological replicate 2 and not replicate 1 of the B-cell sample, nevertheless the authors still plot the half-life of Nup43 in Figure 7A which is misleading. The authors should clarify this inconsistency and describe their approach more accurately.

(6) The manuscript title is too general and needs revision. The authors do not provide sufficient evidence that the observed turnover rates in four non-dividing cell types are generalizable to all non-dividing cell types. In fact, the findings of Toyama et al. strongly dispute this. While the authors may be correct, they have to provide additional evidence that their conclusions are indeed valid.

(7) For Figure 7A, the authors need to cite all relevant literature, in addition to Kosinski et al., they also need to cite Lin et al. *Science*, 2016, which also reported the NPC architecture. The Beck and Hoelz papers appeared back-to-back and should be cited together.

(8) The authors should expand the discussion on the NPC turnover in light of the recent NPC biogenesis findings by the Beck and Ellenberg groups (Hamphoelz et al., *Cell*, 2016; Otsuka et al., *eLife*, 2016).

Reviewer #2 (Remarks to the Author):

In their manuscript entitled "Faster than expected turnover of nucleoporins in non-dividing cells revealed by precise and accurate peptide ion intensity based quantification, Mathieson et al. report improvements to MS1 quantification implemented in the software suite isobarQuant, which they apply to determine protein half-lives in four different non-dividing cell types. They generate a high-quality dataset of protein half-lives for 7291 proteins and find that nucleoporin proteins turn over faster than previously reported in non-dividing brain tissue. While of potential interest, the lack of further experimental validation of the reported turnover rates and the unclear biological implications require additional studies. The authors further claim that mass accuracy is of particular importance for the determination of accurate protein half-lives, however fail to show how their improvement in mass accuracy impacts accurate protein half-life determination through validation with orthogonal methods or by determining contributions of other error sources. While the reviewer agrees that mass accuracy is important, other error sources are possible and in particular for the determination of long half-lives (> 500 hours) using relatively short time points, minor errors will dramatically amplify.

The reported advancements in MS1 quantification are of potential interest to the mass spectrometry community, in particular since distributed as open source software, and refocussing of the manuscript should be considered. However, at this point the authors lack comparison of their MS1 quantification to widely used software packages such as Proteome Discoverer (Thermo) or MaxQuant, which have implemented robust MS1 quantification routines. From the reviewer's perspective, such a comparison is required to justify publication of these otherwise incremental advances to existing software/methodology and should be straight forward given the authors expertise.

Major points:

The authors report an interesting approach by using exact isotope envelope to improve quantification compared to the commonly used averagine model. However, it remains to be shown what impact their approach has compared to existing implementations. The authors show only comparison to a averagine model implemented in isobarQuant.

To make a statement about the improvement in quantification accuracy the authors should at least:

- i. Consider an alternative approach to demonstrate the superior performance of their quantitation approach (other than the protein half-live dataset). For example mix of reference proteomes at given ratios, which allows more direct evaluation of quantification accuracy. The use of a pulsed-SILAC experiment to assess accuracy of quantification appears sub-optimal given the lack of good reference.
- ii. Compare to commonly used MS1 quantification algorithms as implemented in Proteome Discoverer or MaxQuant.
- iii. Estimate systematic/stochastic error and compare between exact model implemented in isobarQuant and existing software solutions.

The authors further introduce novel quality scores based on the fit of the envelope and prior ion ratio, as well as an estimation of co-eluted peptide contamination based on the prior ion ratio. Through combining these improvements the authors seek more accurate MS1 quantification, however fail to show how these compare to commonly used solutions. The authors should provide such comparison as outlined above. Some suggestions:

- i. Compare quantification using the reported improvements to commonly used packages. Compare using different quality metrics such as stochastic error of quantification.
- ii. How does the estimation of co-eluted peptides using prior ion ratio compare to other mechanisms used to detect co-eluted peptides? This should be better discussed.

The authors provide a discussion of how to deal with missing values, the innovation is not obvious to the reviewer. Not inferring data seems a common practice in the field if high data quality is required. The authors should better explain why this is different to common practice.

The understanding of proteome dynamics is an important aspect of better understanding cell biology, and protein half-lives are an important quantity in proteome dynamics. The authors generate an accurate dataset of protein half-lives for 7821 proteins in several types of non-dividing cells, including some greater than 1000 hours. They conclude that proteins of the nuclear core complex exhibit faster turnover than previously reported. While of potential interest, it remains unclear what the biological implications/context of these findings are. Given the discrepancy with previously determined half-lives of nuclear pore proteins, the authors should also provide orthogonal methods to validate some representative half-lives using orthogonal methods such as S35 pulse-chase experiments. The longer half-lives were arguably found in brain tissue, however, given the potential error sources of protein half-live determination using mass spectrometry we feel that validation would be required.

Additional points:

1. It is unclear what the key message/focus of the manuscript is: Better quantitation or faster than expected turnover of NUP proteins. This should be clarified and the manuscript should be refocused.

2. The authors should provide more data/discussion on the influence of various error sources in protein half-life determination.

Reviewers' comments:

Reviewer #1 (Remarks to the Author):

Mathieson et al. describe the development of novel procedures to accurately determine protein half-lives, using a coupled SILAC and mass spectrometry approach. The authors describe substantially higher cellular turnover rates for components of the nuclear pore complex than in a previous report by Toyama et al., which employed rat brain tissue and concluded that nucleoporins are among the longest-lived proteins. Whereas most components of the NPC inner ring complex and the Y-shaped Nup84/Nup107 complex display similarly long half-lives, nucleoporins comprising the Nup358 and Nup214 complexes are rapidly turned over. The authors conclude that the varying turnover rates observed are cell type specific, suggesting different mechanisms for protein turnover within the nuclear pore complex. The reported findings are of potential interest to the readership of Nature Communications. However, the current manuscript requires substantial additional experimental validation of their new mass spec method to provide satisfactory evidence for the physiological relevance expanded upon in the conclusions. This is crucial as the implications of multiple modes of nucleoporin turnover in pre-formed pores would be of considerable interest within the field and undoubtedly will attract significant scrutiny. However, any conclusion regarding the longevity of individual nucleoporins present in otherwise long-lived complexes requires substantial evidence, which is currently insufficient, as this paper may act as the starting point towards uncovering novel machineries that regulate nuclear pore complex turnover.

Major points

(1) Large stretches of the manuscript are extremely technical and require extensive knowledge of mass spectrometry. The authors should revise the main text for increased clarity, aimed towards the broad readership of Nature Communications. In fact, a good case can be made to break up the manuscript into two parts. The authors could publish their novel approach in a specialized proteomics or mass spectrometry journal and the application of the method in a more specialized cell biology journal. As it stands the current manuscript is inaccessible to the broad readership of Nature Communications, who will be hard pressed to discern the limitations of both the methodology and results presented.

Answer: We have shortened and described in a much less technical way the methodology and moved the technical figures to the supplement. We have expanded and focused on the biological findings. In particular, we now perform a proteome wide analysis of protein complex turnover and show that there is a significant statistical enrichment in co-turnover behavior of protein complex subunits. For the case of the proteasome we show that the turnover also reflects complex architecture, with the core particle subunits having a statistically significant difference in its turnover compared to the regulatory particle subunits. For the nuclear pore there is a significant difference in turnover for scaffold and peripheral nucleoporins.

(2) The authors report dramatically different nucleoporin turnover rates compared to the findings of the Hetzer group (Toyama et al., Cell, 2013). Because of these striking differences, it is essential to validate the novel procedures that were employed in the current manuscript. The comparison with previous

findings of protein turnover rates has to be expanded. To establish confidence in the new method, the authors must include analyses of alternative macromolecular complexes with established protein turnover rates.

Answer: We performed additional experiments in mouse embryonic neurons and compared our data to the turnover of histones as well as of the proteasome complex from published work that used different methods. We get an excellent agreement in both cases (less than 2-fold deviation). The histone study used radioactive ¹⁴C labeling (Piha, R.S., M. Cuenod, and H. Waelsch, Metabolism of histones of brain and liver. *J Biol Chem*, 1966. 241(10): p. 2397-404.), while the study where the proteasome turnover was measured (Price, J.C., et al., Analysis of proteome dynamics in the mouse brain. *Proc Natl Acad Sci U S A*, 2010. 107(32): p. 14508-13.) used ¹⁵N labeling. Thus the results obtained with our pulsed SILAC labeling agree with previous *in vivo* measurements in the brain performed using different labeling techniques.

(3) Toyama et al. previously reported that nucleoporins are particularly long-lived in rat brain tissue. The current manuscript now describes a relatively fast turnover of nucleoporins in other non-dividing cells, which is in stark contrast to the previous findings. Once the validity of the procedure is established (see above), a direct comparison to the findings of Toyama et al. is required to address this discrepancy. The reviewer is aware that Toyama et al. sacrificed rats up to 12 months post-chase, which is not a feasible timeframe for a revision. However, the authors could instead use primary rat neuronal cultures for coupled SILAC and mass spectrometry, providing a biologically relevant comparison to the previous findings. Additionally, the NPC analysis should also be comprehensive with the turnover rates of all ~35 nucleoporins presented, not just the turnover rates of a subset.

Answer: As suggested by the reviewer, we performed additional experiments in cultured mouse neurons and found similar turnover rates as compared to the other cell lines analyzed in this study. There are still a number of reasons of both biological and technical nature, why the data obtained from tissue are different. We included an entire paragraph into the discussion to make this as transparent as possible. We primarily want to point out that the extremely long half-life of NPCs previously observed in brain tissue cannot be considered a general phenomenon in all non-dividing cells and we believe we have generated sufficient evidence to claim this. We do not want to prove previous data wrong. As the reviewer rightfully points out this would require tissue measurements which are beyond the scope of the present study.

(4) In Figure 6A, the first biological replicate of the NK and Monocyte samples displays consistently higher protein turnover rates compared to the second biological replicate. The authors should comment on the occurrence of these differences. Moreover, a third biological replicate should be considered to further strengthen the data.

Answer: Thank you for pointing that out. In general half-life measurements are considered to be of high quality if there is less than two-fold difference between replicates. Analyzing the supplement data from the most high profile study in dividing cells to date, Schwanhäusser et al *Nature* 2011, one finds that 85% of all half-lives are within two fold from each other in the two replicates that were performed. In our high quality dataset more than 97% of half-lives are within 2-fold between replicates (99%, 98%,

98%, 96%, and 99% for B cells, hepatocytes, monocytes, NK-cells, and mouse neurons, respectively). Looking at all acquired data without any filtering criteria we get the following numbers, 90% across whole dataset, (93%, 88%, 92%, 85%, 97% for B cells, hepatocytes, monocytes, NK-cells, and mouse neurons, respectively). Thus the quality of this most comprehensive dataset to date of non-dividing cells is as high and higher than that of the most cited study so far on dividing cells.

(5) For several nucleoporins, turnover rates were determined in some cell types but not in others, e.g. Nup37 and Aladin. The authors should justify these differences in detail. Furthermore, the legend to Figure 7A states that protein half-lives were averaged over biological replicates. However, in Supplementary Table 3, the half-life of Nup43 was determined only in biological replicate 2 and not replicate 1 of the B-cell sample, nevertheless the authors still plot the half-life of Nup43 in Figure 7A which is misleading. The authors should clarify this inconsistency and describe their approach more accurately.

Answer: Our study is based on shotgun proteomics, which has a stochastic component. Therefore, it is not the exact same set of proteins quantified in all replicates and cell lines, although they largely overlap. We now make clear in the figure legend that for some nups only one replicate was present. This is now also clear from the supplementary table.

(6) The manuscript title is too general and needs revision. The authors do not provide sufficient evidence that the observed turnover rates in four non-dividing cell types are generalizable to all non-dividing cell types. In fact, the findings of Toyama et al. strongly dispute this. While the authors may be correct, they have to provide additional evidence that their conclusions are indeed valid.

Answer: We agree with this as pointed out in our response to point 3) above. We changed the title of the manuscript as follows:

“Accurate protein half-life determination by mass spectrometry reveals substantial turnover of nucleoporins in primary cells”

(7) For Figure 7A, the authors need to cite all relevant literature, in addition to Kosinski et al., they also need to cite Lin et al. Science, 2016, which also reported the NPC architecture. The Beck and Hoelz papers appeared back-to-back and should be cited together.

Answer: This is a valid point. We apologize for not referencing the Lin paper before, and do so now.

(8) The authors should expand the discussion on the NPC turnover in light of the recent NPC biogenesis findings by the Beck and Ellenberg groups (Hamphoelz et al., Cell, 2016; Otsuka et al., eLife, 2016).

Answer: We agree that the discussion of the physiological relevance of our data had to be expanded and we have done so, although not exactly in the way the reviewer had anticipated, for the following reason: It is tricky to directly relate nucleoporin half-lives in non-dividing cells to knowledge about NPC biogenesis in dividing cells – because the latter continuously synthesize nucleoporins to double the number of NPCs once per cell cycle. We have therefore focused the discussion on other aspects that we

consider interesting, in very brief: that subcomplexes have slightly different turnover times, that peripheral subunits are more short lived, and that multiple compositional populations (in case of the immunoproteasome) might explain our data.

Reviewer #2 (Remarks to the Author):

In their manuscript entitled “Faster than expected turnover of nucleoporins in non-dividing cells revealed by precise and accurate peptide ion intensity based quantification, Mathieson et al. report improvements to MS1 quantification implemented in the software suite isobarQuant, which they apply to determine protein half-lives in four different non-dividing cell types. They generate a high-quality dataset of protein half-lives for 7291 proteins and find that nucleoporin proteins turn over faster than previously reported in non-dividing brain tissue. While of potential interest, the lack of further experimental validation of the reported turnover rates and the unclear biological implications require additional studies. The authors further claim that mass accuracy is of particular importance for the determination of accurate protein half-lives, however fail to show how their improvement in mass accuracy impacts accurate protein half-life determination through validation with orthogonal methods or by determining contributions of other error sources. While the reviewer agrees that mass accuracy is important, other error sources are possible and in particular for the determination of long half-lives (> 500 hours) using relatively short time points, minor errors will dramatically amplify.

The reported advancements in MS1 quantification are of potential interest to the mass spectrometry community, in particular since distributed as open source software, and refocussing of the manuscript should be considered. However, at this point the authors lack comparison of their MS1 quantification to widely used software packages such as Proteome Discoverer (Thermo) or MaxQuant, which have implemented robust MS1 quantification routines. From the reviewer’s perspective, such a comparison is required to justify publication of these otherwise incremental advances to existing software/methodology and should be straight forward given the authors expertise.

Answer: We thank the reviewer for the insightful comments. Following the suggestions of reviewer one, we have now refocused the manuscript on the biological finding, and have moved the technical aspects to the supplement. Nevertheless, we address the technical comments of the reviewer and demonstrate that IsobarQuant is indeed capable of more accurate quantification of pronounced fold-changes even under very challenging, complex sample conditions.

Major points:

The authors report an interesting approach by using exact isotope envelope to improve quantification compared to the commonly used averagine model. However, it remains to be shown what impact their approach has compared to existing implementations. The authors show only comparison to a averagine model implemented in isobarQuant.

To make a statement about the improvement in quantification accuracy the authors should at least:

i. Consider an alternative approach to demonstrate the superior performance of their quantitation approach (other than the protein half-live dataset). For example mix of reference proteomes at given ratios, which allows more direct evaluation of quantification accuracy. The use of a pulsed-SILAC experiment to assess accuracy of quantification appears sub-optimal given the lack of good reference.

Answer: We followed the reviewer's advice and created a very challenging test for MS1 quantification for IsobarQuant and MaxQuant. We mixed light and heavy SILAC labelled cells at different ratios (1-1, 1:9, 1:49), and analyzed the lysed and digested sample without any pre-fractionation thus creating a particularly demanding task for accurate quantification. We analyzed the resulting MS data, both with IsobarQuant and MaxQuant, and can now convincingly show that IsobarQuant performs significantly better than MaxQuant on this challenging complex dataset (Supplementary Figure 4)

ii. Compare to commonly used MS1 quantification algorithms as implemented in Proteome Discoverer or MaxQuant.

Answer: Same answer as above:

We followed the reviewer's advice and created a very challenging test for MS1 quantification for IsobarQuant and MaxQuant. We mixed light and heavy SILAC labelled cells at different ratios (1-1, 1:9, 1:49), and analyzed the lysed and digested sample without any pre-fractionation thus creating a particularly demanding task for accurate quantification. We analyzed the resulting MS data, both with IsobarQuant and MaxQuant, and can now convincingly show that IsobarQuant performs significantly better than MaxQuant on this challenging complex dataset (Supplementary Figure 4)

iii. Estimate systematic/stochastic error and compare between exact model implemented in isobarQuant and existing software solutions.

Answer: Supplementary Figure 4 clearly illustrates that we observe more accurate and precise quantification even for pronounced fold changes than MaxQuant.

The authors further introduce novel quality scores based on the fit of the envelope and prior ion ratio, as well as an estimation of co-eluted peptide contamination based on the prior ion ratio. Through combining these improvements the authors seek more accurate MS1 quantification, however fail to show how these compare to commonly used solutions. The authors should provide such comparison as outlined above. Some suggestions:

i. Compare quantification using the reported improvements to commonly used packages. Compare using different quality metrics such as stochastic error of quantification.

Answer: please see the response to above

ii. How does the estimation of co-eluted peptides using prior ion ratio compare to other mechanisms used to detect co-eluted peptides? This should be better discussed.

Answer: Thank you for pointing that out. So far no one to our knowledge focused on detecting co-eluting peptides which have isotopes of nearly identical m/z as the target precursor. However general detection of co-eluting peptides has been performed before, and we now cite these papers.

The authors provide a discussion of how to deal with missing values, the innovation is not obvious to the reviewer. Not inferring data seems a common practice in the field if high data quality is required. The authors should better explain why this is different to common practice.

Answer: Thank you for pointing this out. Indeed we now tone this part down significantly and only describe the exact way of how we deal with the missing values in the methods section.

The understanding of proteome dynamics is an important aspect of better understanding cell biology, and protein half-lives are an important quantity in proteome dynamics. The authors generate an accurate dataset of protein half-lives for 7821 proteins in several types of non-dividing cells, including some greater than 1000 hours. They conclude that proteins of the nuclear core complex exhibit faster turnover than previously reported. While of potential interest, it remains unclear what the biological implications/context of these findings are. Given the discrepancy with previously determined half-lives of nuclear pore proteins, the authors should also provide orthogonal methods to validate some representative half-lives using orthogonal methods such as S35 pulse-chase experiments. The longer half-lives were arguably found in brain tissue, however, given the potential error sources of protein half-life determination using mass spectrometry we feel that validation would be required.

Answer: We performed additional experiments in mouse embryonic neurons and compared our data to the turnover of histones as well as of the proteasome complex from published work that used different methods. We get an excellent agreement in both cases (less than 2-fold deviation). The histone study used radioactive ¹⁴C labeling (Piha, R.S., M. Cuenod, and H. Waelsch, Metabolism of histones of brain and liver. J Biol Chem, 1966. 241(10): p. 2397-404.), while the study where the proteasome turnover was measured (Price, J.C., et al., Analysis of proteome dynamics in the mouse brain. Proc Natl Acad Sci U S A, 2010. 107(32): p. 14508-13.) used ¹⁵N labeling. Thus the results obtained with our pulsed SILAC labeling agree with previous *in vivo* measurements in the brain performed using different labeling techniques.

Additional points:

1. It is unclear what the key message/focus of the manuscript is: Better quantitation or faster than expected turnover of NUP proteins. This should be clarified and the manuscript should be refocused.

Answer: We have shortened and described in a much less technical way the methodology and moved the technical figures to the supplement. We have expanded and focused on the biological findings. In particular, we now perform a proteome wide analysis of protein complex turnover and show that there is a significant statistical enrichment in co-turnover behavior of protein complex subunits. For the case of the proteasome we show that the turnover also reflects complex architecture, with the core particle subunits having a statistically significant difference in its turnover compared to the regulatory particle subunits.

2. The authors should provide more data/discussion on the influence of various error sources in protein half-life determination.

Answer: In addition to the technical aspects of quantification accuracy we now also discuss the possible effect of recycling of amino acids on half-life determination, in particular in an *in vivo* setting:

“A potential source of inaccuracy for half-life determination is the extent of recycling of the endogenous amino acids in the biological system where they are supposed to be replaced by the isotopically encoded counterparts. This might lead to overestimation of protein half-lives. The effect could be more pronounced *in vivo*, in particular when measuring turnover in organs such as the brain where pulsed amino acids have to traverse several tissues, as well as the blood brain barrier and might exchange with non-labeled amino acids through diverse metabolic and catabolic processes before arriving at the desired destination[18]. The extent to which this effect elevates *in vivo* measurements or how quickly the animal tissues are saturated after the pulse is hard to estimate. Although this effect might elevate absolute values, it is difficult to conceive how it would affect the relative quantification of different complexes to each other. The half-lives measured in non-dividing cells *ex vivo* should yield more accurate values at the cost of losing the endogenous *in vivo* context, which is of course also important.”

Reviewers' comments:

Reviewer #1 (Remarks to the Author):

Mathieson et al. radically revised their manuscript and included all requested experiments and analyses. Specifically, the authors performed a protein turnover analysis in cultured mouse neurons and observed similar protein turnover rates as shown for the four cell types included in the original manuscript. Additionally, the authors carried out a detailed analysis of 26S proteasome turnover rates and compared their results to previously published data. Their analysis agrees convincingly with previously published turnover rates of proteasome subunits and establishes strong confidence in their method. Moreover, the authors drastically improved the accessibility of their manuscript by limiting the technical details to an essential minimum. The quality of the manuscript has substantially improved and now presents the obtained results in a clear manner that is fully accessible to a broad readership.

Overall, the manuscript has the technical quality and broad impact for publication in Nature Communications, but requires further major text revision as detailed below.

The discrepancy between the here reported relatively fast nucleoporin turnover rates *ex vivo* and previous *in vivo* results from the Hetzer group remain unresolved. The authors provide an explanation that *in vivo* protein turnover measurements may be affected by labeled amino acids traversing several tissues before reaching the brain and by recycling of endogenous amino acids that affect the replacement with labeled amino acids. This argument seems flawed, as the authors report less than a 2-fold half-life difference between the 26S proteasome in mouse brain (192 hours, Price et al. 2010, PNAS) and their mouse neuron data (111 hours), but use this argument to explain the drastic differences observed for the NPC.

In the absence of a plausible explanation for the observed differences, the authors should focus on the description and interpretation of the presented data and mention the difference with the Hetzer group as a single point in the discussion. Consequently, all comments on this discrepancy should be removed from the abstract, throughout the manuscript, and from the title, which seems unnecessarily provocative. The slightly aggressive tone should be avoided as it distracts from the otherwise beautiful new method the authors developed.

Reviewer #2 (Remarks to the Author):

In response to reviewer #1, in their revised manuscript Mathieson et al. refocused from technical details on the novel MS1 quantification approach to the biological findings. While this reviewer is not an expert in the biology of the nuclear pore, the novel finding that some nuclear pore proteins turn over faster than previously reported and distinct from other components of the nuclear pore seems not sufficient to merit publication in this journal. The authors do not show or propose a biological function for these observations or a molecular mechanism/pathway, which makes this an interesting observation but not more.

The authors have addressed concerns about accuracy of their approach by comparing to reference data of histones and proteasome.

In addition to the biological findings on the nuclear pore, the authors perform analysis of protein complexes using their extensive dataset of protein half-lives. While such an analysis could have the potential to unravel novel architectural features, and biology, the authors only perform a very superficial analysis that is confirmative of what has been previously known (long half-life of

histones, similar turnover rates for members of a complex).

Also, while the authors state to have moved the novel approach to MS1 quantification (which I still consider of considerable interest if properly validated against other methods) to the methods/supplement, to me the manuscript still reads very method heavy and the novel biology on the nuclear pore is limited to three paragraphs and a single Figure. I believe the authors would benefit from clearly deciding on a focus for their manuscript (nuclear pore, MS1 quantification method/software or systems analysis of turnover rates in complexes), and consider splitting up the manuscript as suggested by reviewer #1.

Major points:

The use of a challenging test set for MS1 quantification (now added to the manuscript) is the right approach, however, the authors have not really responded to the request to quantitatively compare the novel quantification algorithm to existing standards, which they explain by now focusing on the biology.

In their response they state "can now convincingly show that IsobarQuant performs significantly better than MaxQuant on this challenging complex dataset (Suppl. Figure 4)". It is unclear by what metric they judge this improvement.

A) They should provide the exact settings used in MQ and explain why. It would be good to include at least the most commonly used platforms MQ and PD.

B) Explain how they assess quality? Right now, if we look at mean log₂ ratio (which they report in Supp. Fig 4) it is unclear how IBQ outperforms (for 1:1 the theoretical log₂ ratio is 0 and both, ibq and mq return -0.1; for 1:9 the theoretical log₂ ratio is -3.3 and they report -3.3 for mq and -3.2 for ibq; for 1:49 the theoretical log₂ ratio is -5.6 and they report -5.5/5.4 for mq and -5.3/-5.4 for ibq). While the significance of these differences is questionable, if anything mq would be more accurately representing the theoretical mean.

C) The histograms of ratios qualitatively show a disadvantage of re-quantify in MQ but to what extent this impacts on overall dataset quality or is influenced by specific settings in MQ is not shown. One has to consider that in a normal dataset only the most extreme ratios (1:49) will be affected by this, while the majority is likely to be more narrow.

In Figure 3 the authors now present a global analysis of turnover rate correlation within complexes vs. random proteins, which shows an expected behavior, however, it is unclear how exactly this was done and how significant the findings are. It would be interesting to see what happens if the authors cluster data by other factors such as cellular localization or GO terms. Also unclear how the authors correct for complex size? Does their random control group have the same distribution of in terms of number of proteins/group? More details are required to justify more than a supplementary figure.

The authors should discuss how their analysis in Figure 3 fits to the message on differential turnover within complexes.

The observation that the turnover rates for the proteasome core and lid cluster together is a very nice validation for the robustness of their data, however, not much more and somewhat out of context for a manuscript now focusing on the nuclear core complex. This should be moved to the supplement.

The authors need to better discuss what novel implications their finding on turnover rates for the nuclear pore complex has.

Minor points:

It is unclear how the authors come to their numbers of half-lives for 9699 proteins, which they prominently cite in the abstract and throughout the text. Looking over the datasets the average number of accurately quantified half-lives appears to be in the order of ~ 4000 . I don't think it is necessary to overstate these numbers and to me citing 9699 half-lives the way it is done implies a dataset with 9699 half-lives from one cell type across multiple replicates.

We thank the reviewers for their appreciation of the manuscript and their constructive comments. We address all comments point by point below.

Reviewer #1 (Remarks to the Author):

Mathieson et al. radically revised their manuscript and included all requested experiments and analyses. Specifically, the authors performed a protein turnover analysis in cultured mouse neurons and observed similar protein turnover rates as shown for the four cell types included in the original manuscript. Additionally, the authors carried out a detailed analysis of 26S proteasome turnover rates and compared their results to previously published data. Their analysis agrees convincingly with previously published turnover rates of proteasome subunits and establishes strong confidence in their method. Moreover, the authors drastically improved the accessibility of their manuscript by limiting the technical details to an essential minimum. The quality of the manuscript has substantially improved and now presents the obtained results in a clear manner that is fully accessible to a broad readership. Overall, the manuscript has the technical quality and broad impact for publication in Nature Communications, but requires further major text revision as detailed below.

The discrepancy between the here reported relatively fast nucleoporin turnover rates *ex vivo* and previous *in vivo* results from the Hetzer group remain unresolved. The authors provide an explanation that *in vivo* protein turnover measurements may be affected by labeled amino acids traversing several tissues before reaching the brain and by recycling of endogenous amino acids that affect the replacement with labeled amino acids. This argument seems flawed, as the authors report less than a 2-fold half-life difference between the 26S proteasome in mouse brain (192 hours, Price et al. 2010, PNAS) and their mouse neuron data (111 hours), but use this argument to explain the drastic differences observed for the NPC.

In the absence of a plausible explanation for the observed differences, the authors should focus on the description and interpretation of the presented data and mention the difference with the Hetzer group as a single point in the discussion. Consequently, all comments on this discrepancy should be removed from the abstract, throughout the manuscript, and from the title, which seems unnecessarily provocative. The slightly aggressive tone should be avoided as it distracts from the otherwise beautiful new method the authors developed.

We want to thank the reviewer for this overall very positive assessment.

We agree that our data do not ultimately proof or disproof previous *in vivo* work of the Hetzer Lab, and to challenge this data would be beyond the scope of our manuscript. Nevertheless, we can thoroughly show that in various non-dividing cell types, nucleoporins turn over rather quickly. The conclusion that the previously observed, rather slow *in vivo* turn-over is not a general attribute of non-dividing cells is thus valid, novel and interesting.

We agree with the reviewer that this has to be explained better, and by no means we meant to be aggressive. As suggested by the reviewer, we made various textual edits, including changes of title and abstract to focus the key messages on the facts. The respective discussion section for example, we have re-phrased as follows:

“The technical details but also the biological context of both experiments are very difficult to compare and either of which have their benefits. The half-lives measured in non-dividing cells in vitro should yield more accurate values at the cost of losing the endogenous in vivo context, which is of course also important. We can nevertheless conclude that the very slow NPC turnover is not a general phenomenon of all non-dividing cells.”

Reviewer #2 (Remarks to the Author):

In response to reviewer #1, in their revised manuscript Mathieson et al. refocused from technical details on the novel MS1 quantification approach to the biological findings. While this reviewer is not an expert in the biology of the nuclear pore, the novel finding that some nuclear pore proteins turn over faster than previously reported and distinct from other components of the nuclear pore seems not sufficient to merit publication in this journal. The authors do not show or propose a biological function for these observations or a molecular mechanism/pathway, which makes this an interesting observation but not more.

The authors have addressed concerns about accuracy of their approach by comparing to reference data of histones and proteasome.

In addition to the biological findings on the nuclear pore, the authors perform analysis of protein complexes using their extensive dataset of protein half-lives. While such an analysis could have the potential to unravel novel architectural features, and biology, the authors only perform a very superficial analysis that is confirmative of what has been previously known (long half-life of histones, similar turnover rates for members of a complex).

We agree that our data set would in principle allow an in-depth analysis of protein complexes, however, this would require a considerable amount of time and also a different type of expertise. We hope that the reviewer agrees that this would be beyond the scope of the present manuscript. We want to stress that our findings regarding the nuclear pore are, in our view, a very interesting biological result (see also response to individual point below). These results exemplify the power of our method and the usefulness of our data set as a resource.

Also, while the authors state to have moved the novel approach to MS1 quantification (which I still consider of considerable interest if properly validated against other methods) to the methods/supplement, to me the manuscript still reads very method heavy and the novel biology on the nuclear pore is limited to three paragraphs and a single Figure. I believe the authors would benefit from clearly deciding on a focus for their manuscript (nuclear pore, MS1 quantification method/software or systems analysis of turnover rates in complexes), and consider splitting up the manuscript as suggested by reviewer #1.

We want thank the reviewer for this comment, which helped us to further improve the manuscript. Following the feedback from Reviewer 1 as well as a discussion with the editor, we have kept the current structure of the manuscript, but have made an effort to make the technical part of the results section more appealing and understandable for a broader audience.

Major points:

The use of a challenging test set for MS1 quantification (now added to the manuscript) is the right approach, however, the authors have not really responded to the request to quantitatively compare the novel quantification algorithm to existing standards, which they explain by now focusing on the biology. In their response they state “can now convincingly show that IsobarQuant performs significantly better than MaxQuant on this challenging complex dataset (Suppl. Figure 4)”. It is unclear by what metric they judge this improvement.

A) They should provide the exact settings used in MQ and explain why. It would be good to include at least the most commonly used platforms MQ and PD.

Reply: We had provided the settings used for MaxQuant in the methods section. To make this more transparent, we now added a heading “Comparison between IsobarQuant and MaxQuant” and we provide all the details on all settings of MaxQuant in the methods section as well as in a new supplementary table 2. The settings we used are the standard recommended settings that have been used in hundreds of studies. We choose to restrict our comparison to MaxQuant since we use it a lot and are highly familiar with it, and thus can be certain that we are making a fair comparison.

B) Explain how they assess quality? Right now, if we look at mean log₂ ratio (which they report in Suppl. Fig 4) it is unclear how IBQ outperforms (for 1:1 the theoretical log₂ ratio is 0 and both, ibq and mq return -0.1; for 1:9 the theoretical log₂ ratio is -3.3 and they report -3.3 for mq and -3.2 for ibq; for 1:49 the theoretical log₂ ratio is -5.6 and they report -5.5/5.4 for mq and -5.3/-5.4 for ibq). While the significance of these differences is questionable, if anything mq would be more accurately representing the theoretical mean.

Reply: The reviewer is correct in that the mode values for MaxQuant and isobarQuant are very close. Our conclusion is based on the fact that MaxQuant in its sensitive setting (with the re-quantify function switched on) accumulates a large number of poorly quantified peptides - which the reviewer also states in the comment below. The isobarQuant strategy for going back and quantifying peptides after peptide identification, is conceptually similar to MaxQuant’s re-quantify function and we show that it performs better. This becomes very clear when looking at the median fold-changes instead of the mode, which we now also add to all plots. It is apparent that for the 1:49 sample MaxQuant with re-quantify function switched off provides the most accurate median, -5.5, but at the cost of very heavy losses in peptide numbers. MaxQuant with re-quantify switched on has excellent coverage but the median is very strongly off, -4.0. isobarQuant without any filtering, already improves the median substantially, -4.8, and isobarQuant with filtering significantly improves it further, -5.2.

For the 1:9 sample the difference in medians is absent, but a clear reduction in the amount of poorly quantified peptides when using isobarQuant is clearly observed.

Thus, by adjusting the filtering criteria in isobarQuant we are able to achieve a good balance between peptide coverage and accuracy. The tunable filter settings according to the developed parameters enable more flexibility than the binary setting in MaxQuant and in principle give us the possibility to rank the peptides according to expected quantification quality. More detailed investigations in this

direction will indeed be a subject of a very technical paper in the future. For now, we just want to state in a clear way that isobarQuant provides the possibility to increase peptide coverage of low fold-change peptides by increasing the number of good quality peptides, thus enabling more sensitive quantification of proteins with pronounced fold-changes.

We would also like thank the reviewer for stating that he considers these innovations to be of considerable interest.

C) The histograms of ratios qualitatively show a disadvantage of re-quantify in MQ but to what extent this impacts on overall dataset quality or is influenced by specific settings in MQ is not shown. One has to consider that in a normal dataset only the most extreme ratios (1:49) will be affected by this, while the majority is likely to be more narrow.

As stated above, we now explicitly state the MaxQuant settings used. These are standard settings that can be found in hundreds of publications. We show that isobarQuant's post identification quantification strategy is superior to the conceptually similar MaxQuant's re-quantify strategy as the reviewer correctly points out in the comment, and as explained in the response above. The reviewer is correct that for most datasets this will not be an issue when performing quantification with MaxQuant. It can be an issue in studies similar to ours where robust quantification of pronounced fold changes is desired for some proteins. This is the point we want to make. The reviewer is entirely correct that this will not be a problem in most normal datasets. We apologize for stating this too bluntly in the previous response letter. Like the reviewer, the last thing we want is for people to draw the conclusion that MaxQuant might perform badly on normal proteomic datasets. We therefore use careful phrasing to indicate that isobarQuant performs better than MaxQuant only in special settings when for some proteins quantification of pronounced fold changes is desired.

In Figure 3 the authors now present a global analysis of turnover rate correlation within complexes vs. random proteins, which shows an expected behavior, however, it is unclear how exactly this was done and how significant the findings are. It would be interesting to see what happens if the authors cluster data by other factors such as cellular localization or GO terms. Also unclear how the authors correct for complex size? Does their random control group have the same distribution of in terms of number of proteins/group? More details are required to justify more than a supplementary figure.

Reply: We now add as a supplementary figure prior to complex analysis the analysis of turnover in different cellular compartments. We believe this adds value to the manuscript and we thank the reviewer for this suggestion. From the manuscript:

"We also assessed the turnover behavior within the different cellular compartments, (Supplementary Fig. 6). Within each compartment a broad range of protein half-lives was observed and there is a large overlap in turnover behavior between all compartments. However, significant differences in general turnover behavior are observed between the different compartments, although the effect size is small, (Supplementary Fig. 6). The most prominent and significant trend is the slower turnover of the mitochondrial proteins, which is present in all cell types. Proteins from the Golgi apparatus and nucleus had reproducibly the slowest turnover, while endoplasmic reticulum and cytoplasmic proteins were located in the middle with close to identical turnover distributions. "

The random group for the complex analysis is created so that the distribution of in terms of number of proteins/group is preserved. We explain that in the method section, so we now make it clearer and also state in the main text. We have now improved the integration of Figure 3 into the manuscript flow:

“We calculated the standard deviations of the half-life values of proteins that are subunits of the same annotated complex, and compared those to the standard deviations obtained for the same complexes after the subunits were reshuffled across all complexes (see methods). A clear and significant trend (p -value <0.001) for a more coherent half-life distribution of protein subunits within individual complexes becomes apparent for all cell types, Figure 3. The chaperonin complex has the most tightly controlled turnover of the individual subunits in all different cell types. Looking at two larger complexes with more intricate architecture: the nuclear pore complex, (NPC), and the 26S proteasome we observe a much less tightly controlled turnover, except for the 26S proteasome in mouse neurons where the turnover behavior is very coherent.”

The authors should discuss how their analysis in Figure 3 fits to the message on differential turnover within complexes.

Reply: We have now improved the integration of Figure 3 into the manuscript flow, and connect the results from it with the observation for NPC and proteasome:

“We calculated the standard deviations of the half-life values of proteins that are subunits of the same annotated complex, and compared those to the standard deviations obtained for the same complexes after the subunits were reshuffled across all complexes preserving the number of proteins in each complex group (see methods). A clear and significant trend (p -value <0.001) for a more coherent half-life distribution of protein subunits within individual complexes becomes apparent for all cell types, Figure 3. The chaperonin complex has the most tightly controlled turnover of the individual subunits in all different cell types. Looking at two larger complexes with more intricate architecture: the nuclear pore complex, (NPC), and the 26S proteasome we observe a much less tightly controlled turnover, except for the 26S proteasome in mouse neurons where the turnover behavior is very coherent.

Looking in more detail at the 26S proteasome, Figure 4A, Supplemental Table 4, we observe significantly different half-lives of the 20S core complex subunits compared to 19S regulatory complex subunits in all cell types except for mouse embryonic neurons Figure 4A, which explains the more coherent turnover behavior in these cells. Interestingly there is a significant trend for the 20S core complex to be more stable than the 19S regulatory complex in B-cells, monocytes, and NK cells, but a clear and significant opposite behavior is observed for hepatocytes. Clustering of the proteasome subunits according to the similarity between their half-lives across the human cells also leads to distinct separation between the core and regulatory subunits (Figure 4B).

Next, we looked at the NPC”

We now mark the proteasome, NPC, and the chaperonin complex in Figure 3 (we also corrected a minor bug in the Figure, but the conclusions are identical). There is a significant trend for coherent complex turnover, but it is reproducibly more apparent in some complexes than in others, due to both different

turnover of subcomplexes as well as faster turnover of peripheral complex members for the two large complexes that we investigate in detail.

The observation that the turnover rates for the proteasome core and lid cluster together is a very nice validation for the robustness of their data, however, not much more and somewhat out of context for a manuscript now focusing on the nuclear core complex. This should be moved to the supplement.

Reply: We now integrate the proteasome analysis better into the manuscript flow, see above. Thank you for commenting on this, now things fit much better together. The proteasome is one of the few complexes for which previous, independent data exist. We thus would prefer to keep it highlighted in the manuscript.

The authors need to better discuss what novel implications their finding on turnover rates for the nuclear pore complex has.

We agree and have edited the discussion section accordingly to make this more obvious. In very brief, it presently remains debated if cells selectively turn over NPCs because such events are extremely rare in basically any biological system that has been looked at. Specific NPC turn-over pathways have not yet been identified. Toyama et al (Hetzer/Yates Labs) have found that in rat brain, NPC proteins and histones have half-lives of ~6 month, which is extreme as compared to most other proteins and accounts for a quarter of the animal's life span. If this phenomenon can account for non-dividing cells in general, that do not dilute out their NPCs over time by generating two daughter nuclei during each cell division, remains unknown. Our data strongly suggest that NPC proteins do turn-over rather quickly, also in non-dividing cells. This finding suggests the existence of NPC turn-over pathways and is important for the respective scientific field.

Minor points:

It is unclear how the authors come to their numbers of half-lives for 9699 proteins, which they prominently cite in the abstract and throughout the text. Looking over the datasets the average number of accurately quantified half-lives appears to be in the order of ~ 4000. I don't think it is necessary to overstate these numbers and to me citing 9699 half-lives the way it is done implies a dataset with 9699 half-lives from one cell type across multiple replicates.

Reply: 9699 is the number of proteins across all cell types combined. The reviewer is correct that the sentence in the abstract can be misinterpreted we now write more clearly:

"This enabled precise and accurate protein half-life determination ranging from 10 to more than 1000 hours. We achieve good proteomic coverage ranging from four to six thousand proteins in several types

of non-dividing cells, corresponding to a total of 9699 unique proteins over the entire dataset.”. We also make sure that it is clear whenever mentioned in other parts of the manuscript.

Thank you for pointing this out.

REVIEWERS' COMMENTS:

Reviewer #2 (Remarks to the Author):

The authors have addressed most my concerns from the previous rounds of reviews including all technical concerns. Mathieson et al., have further considerably revised their manuscript, and in the current form it is much more accessible to a broad readership and provides a resource for protein half-lives in non-dividing cells of excellent quality. While I still remain skeptical about the level of conceptual advance around nuclear pore complex turnover, the authors have now integrated these findings well with more general conclusions around protein stability in non-dividing cells and the turnover behavior of protein complexes. The balanced discussion of technical advance, the use of large-scale datasets, and the presentation of the exemplary analysis for NPC and Proteasome have significantly improved the manuscript.